# Tunable Ammonia Adsorption within Metal–Organic Frameworks with Different Unsaturated Metal Sites

**DOI:** 10.3390/molecules27227847

**Published:** 2022-11-14

**Authors:** Dongli Zhang, Yujun Shen, Jingtao Ding, Haibin Zhou, Yuehong Zhang, Qikun Feng, Xi Zhang, Kun Chen, Jian Wang, Qiongyi Chen, Yang Zhang, Chaoqun Li

**Affiliations:** 1Academy of Agricultural Planning and Engineering, Key Laboratory of Technologies and Models for Cyclic Utilization from Agricultural Resources, Ministry of Agriculture, Beijing 100125, China; 2School of Advanced Manufacturing, Guangdong University of Technology, Jieyang 515200, China; 3State Key Laboratory of Power Systems, Department of Electrical Engineering, Tsinghua University, Beijing 100084, China; 4College of Materials Science and Technology, Hainan University, Haikou 570228, China

**Keywords:** metal–organic frameworks (MOFs), unsaturated metal sites, ammonia, adsorption capacity, absorption mechanisms

## Abstract

Ammonia (NH_3_) emissions during agricultural production can cause serious consequences on animal and human health, and it is quite vital to develop high-efficiency adsorbents for NH_3_ removal from emission sources or air. Porous metal–organic frameworks (MOFs), as the most promising candidates for the capture of NH_3_, offer a unique solid adsorbent design platform. In this work, a series of MOFs with different metal centers, ZnBTC, FeBTC and CuBTC, were proposed for NH_3_ adsorption. The metal centers of the three MOFs are coordinated in a different manner and can be attacked by NH_3_ with different strengths, resulting in different adsorption capacities of 11.33, 9.5, and 23.88 mmol/g, respectively. In addition, theoretical calculations, powder XRD patterns, FTIR, and BET for the three materials before and after absorption of ammonia were investigated to elucidate their distinctively different ammonia absorption mechanisms. Overall, the study will absolutely provide an important step in designing promising MOFs with appropriate central metals for the capture of NH_3_.

## 1. Introduction

Ammonia (NH_3_), as a highly toxic and corrosive gas, is a rather common pollutant in agricultural production. Additionally, ammonia has a stimulating effect on human upper airways and eyes, even in small concentrations. A large amount of NH_3_ emissions have brought huge hidden dangers to the environment and economy [1,2,3]. Hence, ammonia pollution is such a serious problem that it has aroused concerns from the whole society. Furthermore, the research interests in the treatment of NH_3_ pollution have increased among relevant departments of academia and government [4,5,6]. It is well known that poultry and animal husbandry, fossil fuel combustion, refrigeration industry, fertilizer manufacturing and coke manufacturing could generate a large quantity of NH_3_ [7,8]. The reduction in NH_3_ emissions has been placed in an important position. In addition, due to the dispersion and uncertainty of NH_3_ pollution, the effective mechanism for resolving NH_3_ pollution has not been systematically established. There are two primary types of ammonia adsorption: physisorption and chemisorption. The physisorption is related to the surface area and pore volume, while chemisorption depends on coordinate bonding, including the unsaturated metal sites, functional groups, acidic binding sites, and defective sites. The adsorption mechanism for ammonia adsorption still remains an important issue to be solved both theoretically and practically.

Activated carbon, zeolite molecular sieves and metal oxides, belonging to traditional ammonia adsorption materials, are believed to be a practical solution for ammonia pollution. However, they suffer from low selectivity and low capacity for ammonia [9,10,11,12,13]. Nevertheless, MOFs (metal–organic frameworks), as a typical porous material, were formed by the self-assembly of organic ligands with metal ions or metal clusters and have the advantages of large specific surface area, high porosity, ordered pore structure and modifiable structure [14,15,16,17,18,19]. MOFs are widely used in catalysis, gas storage and separation and molecular recognition because of their merits. In addition, MOFs have been postulated as a promising option for the removal of NH_3_ [20,21,22,23].

Among the widely studied MOFs, ZIF-8, Cu-BTC, UiO and MIL series MOFs materials have been widely employed for ammonia adsorption in previous research works. ZIF-8, as a well-known MOF, can even be mass-produced and put into application properly, but its ammonia adsorption capacity is low [24]. CuBTC, as reported by Peterson et al., has very high ammonia uptake [25] because it could interact strongly with ammonia due to the presence of open metal sites. Moreover, MOFs with the incorporation of different functional groups have also been explored for NH_3_ removal. As reported by Wu et al. [26], the incorporation of biphenyl and bipyridine groups into MOFs UiO-67 and UiO-bpydc could lead to drastically different NH_3_ adsorption properties because the bipyridine moieties can induce flexibility to the framework without significant pore volume alteration. In addition, Chen’s group reported that MIL-100 and MIL-101 have a large NH_3_ uptake of 8 and 10 mmol/g, respectively, and the NH_3_ adsorption capacity could be improved by modified amino functional groups. More importantly, all MOF materials exhibit excellent stability and recycled NH_3_ removal [27]. Although some common MOFs were reported in the field of NH_3_ adsorption [28,29], many other MOFs, such as ZnBTC [30], are rarely involved. Specifically, the effectiveness of MOFs with different open metal sites for ammonia capture was not systematically studied.

In this study, three monometallic adsorbents, ZnBTC, FeBTC and CuBTC, were prepared by hydrothermal methods and their NH_3_ adsorption property has been investigated. The phase structures and adsorption performances of synthesized MBTC materials with different metal central ions were characterized by SEM, BET, XRD and FTIR, respectively. Their main adsorption paths and mechanisms were further explored by DFT calculations, which could provide theoretical support for the preparation of the adsorbents for ammonia in the later stage.

## 2. Experimental Section

### 2.1. Materials and Instruments

All reagents were used as supplied without further purification. The benzene-1,3,5-tricarboxylic acid was obtained from the Sinopharm Chemical Reagent Co., Ltd. (Beijing, China). Zn(NO_3_)_2_·6H_2_O (AR, 99%), Cu(NO_3_)_2_·3H_2_O (AR, 98%) and Fe(NO_3_)_2_·9H_2_O and ZnAc_2_·2H_2_O were purchased from Macklin Co. Ltd. (Shanghai, China). Ethanol and H_2_O were commercially available.

### 2.2. Synthesis of ZnBTC, FeBTC and CuBTC

The synthetic methods of ZnBTC, FeBTC and CuBTC were determined according to the reported literature [28,29,30]. The preparation process comprises the following steps. First, Zn(NO_3_)_2_·6H_2_O (1.8 g), H_3_BTC (0.6 g), and ethyl alcohol-H_2_O (1:1, 60 mL) were well-mixed in a 100 mL vial by constantly stirring for 20 min. Then, the mixture was heated at 120 °C for 16 h. After cooling down to room temperature, the white crystals of Zn-BTC precipitate were filtered and then washed with ethanol and deionized (DI) water several times until the filtrate was neutral. Finally, the Zn-BTC was dried in a vacuum oven at 60 °C for 12 h.

FeBTC and CuBTC were prepared in similar methods [27,31].

### 2.3. Adsorption Experiments

The static capacity sorption analyzer of Bei Shi De (BSD-PSPM) was used to estimate the adsorption uptake of the pure NH_3_. Before testing, the samples were activated at 50 °C for at least 2 h until the mass no longer changed. For the co-adsorption of NH_3_/H_2_O, 0.10 g of adsorbent was in an enclosed area with 20 mL of NH_3_/H_2_O (4:1, *v*:*v*). After the saturated adsorption was reached, the mixture was added to a saturated solution of potassium chloride using a temperature-controlled shaker to extract ammonia nitrogen. The filter liquid was achieved by membrane filtration methods and analyzed by Automatic Discrete Analyzer (SmartChem 140) to evaluate the adsorption uptake of NH_3_.

### 2.4. Characterization

Scanning electron microscopy (SEM) images of the prepared MOFs were obtained by a field-emission scanning (SEM, S-4700). Fourier transform infrared (FT-IR) spectroscopy was carried out by Nicolet 6700FT-IR spectrophotometer. The crystal structure and phase purity of materials were collected on an X-ray diffractometer (Bruker AXS D8-Advance) in 2θ range from 5° to 40° at a scan rate of 10° min^−1^. The specific surface area of the materials in this study was obtained by using a surface and micropore analyzer of BEST Instrument Technology Co., Ltd (Beijing, China). The Brunauer–Emmett–Teller (BET) surface area was obtained using a volumetric method. Using the Gaussian09 (Revision D.01) package [32], the calculations of ZnBTC, FeBTC and CuBTC in this study were carried out by the density functional theory (DFT) method, which employs the B3LYP [33] functional. The basis set was employed, LANL2DZ, for metal elements and 6–31 G(d) [34] for others. The values 2.7 × 10^−4^ eV (energy), 0.05 eV/Å (gradi-ent), and 0.005 Å (displacement) are defined as the parameters of the convergence threshold [35]. The binding energy (Δ*E*) was calculated as follows:

For MOFs + NH_3_ MOFs_(NH_3_)_
Δ*E* = *E*_MOFs(NH_3_)_ − *E*_MOFs_ − *E*_NH_3__(1)
where *E*_MOF_, *E*_NH_3__ and *E*_MOFs(NH_3_)_ represent the energy of the MOF unit, NH_3_, and the MOFs after adsorption of ammonia, respectively.

## 3. Results and Discussion

The crystal morphologies of the as-synthesized materials, ZnBTC, FeBTC and CuBTC, were characterized by SEM. Before observation, the samples were sprayed with Au particles. As illustrated in Figure 1, the particles of the three crystalline materials are not uniform in size, with a particle size distribution of 1–5 μm. The surface morphology of the material is relatively rough. Taking the CuBTC, for example, there are some fixtures on the surface (Figure 1e,f). It is worth noting that the three MOFs bearing different metal centers take on quite different microstructures. In detail, ZnBTC is a rod-like structure with uneven length, while FeBTC and CuBTC exhibit an irregular blocky structure and a sheet-like structure, respectively. These rather distinctive morphology images of the three different materials may be due to their different metal centers involving different coordination patterns between the metal centers and the ligands.

The TGA analysis was adopted to evaluate the thermal stability of the three MOFs. As shown in Figure 2a, the pyrolysis processes of ZnBTC and CuBTC are similar and can be mainly divided into three stages. It can be found that ZnBTC and CuBTC exhibit visible weight loss from 30 to 200 °C (calculated: 6.42% for ZnBTC, 10.02% for CuBTC), which can be ascribed to the desorption of coordinated guest water molecules formed between unsaturated metal sites and oxygen molecule through the high affinity. The weight loss between 300 and 550 °C may be due to the partial decomposition of the ligand, and the final weight loss can be attributed to the carbonization of the sample and the formation of oxides. The weight loss process of FeBTC during pyrolysis is evidently different from those of ZnBTC and CuBTC. The mass loss of FeBTC with the temperature range of 30 to 200 °C originates from the evaporation of adsorbed water (calculated 4.82%), while the weight loss at 200–400 °C may be caused by the partial decomposition of the ligands. The decrease in weight for FeBTC during 400–550 °C should be attributed to the decomposition of some carboxyl groups into graphitized carbon. The final weight loss stage can be attributed to further carbonization of the sample and reduction in iron species. The residual weights of ZnBTC, CuBTC and FeBTC at 700 °C are 40.29%, 38.03% and 30.04%, respectively, indicating that the thermal stability of ZnBTC and CuBTC is superior to that of FeBTC. It is worth noting that the three compounds behave differently in the first weight loss stage within the temperature range from 30 to 200 °C, with a respective weight loss of 6.42% for ZnBTC, 10.02% for CuBTC, 4.82% for FeBTC, suggesting the increasing adsorption capacity of polar water molecules from FeBTC, ZnBTC, and CuBTC. In other words, the interaction between the metal center Cu and polar molecules is the strongest of the three compounds.

Isothermal adsorption and desorption measurements were conducted to characterize parameters such as specific surface area. Before testing, a degassing pretreatment was performed under vacuum. The specific surface area of the three MOFs was determined by the BET and Langmuir methods, respectively. The nitrogen adsorption and desorption isotherms of ZnBTC, FeBTC and CuBTC all show Type I adsorption isotherms (Figure 2b), indicating that the three materials have microporous structures. As can be observed, the adsorption amount of the material rises sharply when the relative pressure is low. This is mainly due to the filling effect of the micropores and the enhancement of the adsorption potential in the micropores, which lead to adsorbate molecules having a strong capture capacity at lower relative pressures. Further, with the increase in relative pressure, the adsorption amount increased slowly and the growth rate was small, indicating that the adsorption almost reached saturation. The adsorption amount increased slowly with relative pressure and may mean it has no potential to rise, as is made evident by the deformed saturation point of FeBTC at 1 bar. When the pressure P/P_o_ of CuBTC and ZnBTC was close to 1, the adsorption curve was upturned and the adsorption capacity increased. The main reason is that the two MOF samples are both nanoparticles, and nitrogen molecules condense on the surfaces of the pore structure formed by the accumulation between particles. It can be calculated from the nitrogen adsorption and desorption curves that the specific surface areas of FeBTC, CuBTC and ZnBTC are 1407.04, 388.68 and 44.47 m^2^/g, respectively.

To elucidate the NH_3_ adsorption properties of the three MOFs, NH_3_ adsorption and desorption isotherms of ZnBTC, FeBTC and CuBTC were performed. As shown in Figure 3a, ZnBTC, FeBTC and CuBTC exhibited distinct NH_3_ uptake amounts of 5.04, 4.33 and 15.35 mmol/g, respectively, at low adsorption pressure (25 mbar). With the increase in pressure, the adsorption capacity of all three materials slowed down in growth. For FeBTC and CuBTC, their adsorption capacity still slightly improved, while that for ZnBTC plateaued in the high-pressure region. The distinctively different performances of the three MOFs may result from the metal sites in MOFs. In addition, the results indicate that adsorption capacity has no apparent relevance to specific surface area and pore size. Finally, the adsorption amount of NH_3_ for ZnBTC, FeBTC and CuBTC reached 11.33, 9.5, and 23.88 mmol/g, respectively, at 1 bar.

During desorption at different pressures, all three materials exhibit apparent lag, suggesting a strong interaction between the open metal sites and NH_3_. As shown in the desorption curves, only a small amount of NH_3_ is desorbed initially, and there still is a large amount of NH_3_ that cannot be completely removed when the pressure drops as low as 0.1 bar. Based on the high absorption amount from the low-pressure region and the fact that a large amount of NH_3_ cannot be desorbed, it can be assumed that NH_3_ is chemically adsorbed in the three MOFs. The distinctive ammonia adsorption performance of CuBTC should originate from the strong interaction between its metal center copper and the polar molecule ammonia, as exemplified by the strong interaction between copper and water detailed above. In particular, the role of unsaturated metal sites dominates, resulting in a rapid increase in the adsorption capacity. Finally, it calls for putting the measures mentioned above into practice, such as ammonia water, and further demonstrates the application of MOFs to absorb NH_3_ in moist conditions. The adsorption amount of NH_3_ was calculated by the desorption of ammonium ions. The adsorption capacity for CuBTC, ZnBTC and FeBTC under the atmosphere of ammonia water was 10.47, 6.03, and 5.39 mmol/g, respectively.

To confirm the crystallinity of ZnBTC, FeBTC and CuBTC before and after NH_3_ adsorption, XRD patterns of the three MOFs are inspected in Figure 4a. The peak positions of different materials are in complete agreement with the simulated data reported previously, confirming that the material has been successfully synthesized [25,30,36]. After the co-adsorption of H_2_O/NH_3_, the PXRD of CuBTC changed significantly, indicating it was transformed into Cu_3_(BTC)_2_(NH_3_)_6_(H_2_O)_3_ [37]. The adsorption peaks of ZnBTC and FeBTC also changed significantly, indicating that the metal centers of ZnBTC and FeBTC may have been coordinated with NH_3_, respectively.

To explore the types of functional groups in these three synthetic materials, FTIR were performed with the range 4000–4500 cm^−1^, and the result is consistent with that reported previously [38]. As shown in Figure 4b, all the three MOFs exhibit similar absorption peaks, indicating the presence of similar functional groups in the three materials. Taking FeBTC, for example, the signal at 3345 cm^−1^ corresponds to the -OH peak on the carboxylic acid ligand, and the absorption peak at 1625 cm^−1^ represents the C=O on the carboxylic acid ligand and water H-O-H vibration peaks. The absorption signals at 1384 and 1618 cm^−1^ are the stretching vibration signals of C-O on the carboxyl group, and the vibrational peaks of C-C on the benzene ring were at 1438 cm^−1^, respectively. The signals at 712 and 761 cm^−1^ are the C-H vibrations on the benzene ring. After NH_3_/H_2_O co-adsorption, a broad absorption peak of NH_3_ at ca. 3300 cm^−1^ appears in the spectrum, indicating that NH_3_ molecules are incorporated into the structure of MBTC. Moreover, free organic ligands in each MOF were observed, suggesting the destruction of the frameworks.

To further understand the adsorption mechanism of the NH_3_ over ZnBTC, FeBTC and CuBTC, this study simulates the microstructure of MOFs adsorption materials. The crystal structures of Zn-BTC, FeBTC and CuBTC were referenced from the Cambridge Crystallographic Data Centre (CCDC) [27,28,29,30,35]. In order to simulate the local environment and reduce the calculation amount, the optimization of MOF structures was essential. The all-atom frameworks were adopted to analyze the canonical clusters of the MOFs. For example, to retain the correct hybridization, the methyl (−CH_3_) groups were used to cut off the dangling bonds from these fragmented clusters.

The adsorption sites, configurations, and adsorption energies of NH_3_ over Zn-BTC, FeBTC and CuBTC are obtained by the calculation. As illustrated in Figure 5, the NH_3_ molecules attacked the metal center of MBTC (M=Cu, Zn, Fe) in different ways. The CuBTC and ZnBTC are attacked relatively easily as they have high polarity, strong coordination ability and simple configurations. Hence, the NH_3_ could attack CuBTC and ZnBTC in rectilinear form. Conversely, FeBTC is not vulnerable to the attack due to the cage structure, which forms a large steric hindrance effect. Overall, the connection between the metals (Cu, Zn, Fe) and BTC ligands was damaged, following the combination of NH_3_ molecules.

Moreover, the binding energy on M sites over ZnBTC, FeBTC and CuBTC are −32.9, −18.7 and −14.7 kcal/mol, respectively, indicating that the ZnBTC owns higher adsorption strength than CuBTC and FeBTC (Table 1). The different trend conforms with the distance from the M sites to NH_3_, which are 2.146, 2.298 and 2.282 Å for ZnBTC, FeBTC and CuBTC, respectively. Our preliminary results suggest that the BET and the distance from the M sites to NH_3_ have a great correlation with adsorption capacity but are not the only factors that decide the adsorption capacity.

## 4. Conclusions

In summary, three metal–organic frameworks (MOFs) with different metal centers, ZnBTC, FeBTC and CuBTC, were designed, synthesized, and characterized by a series of techniques such as XRD, FT-IR. The results illustrate that CuBTC exhibits a rather large ammonia adsorption capacity of 23.88 mmol/g compared with the other two (ZnBTC (11.33 mmol/g) and FeBTC (9.5 mmol/g)). As disclosed by theoretical calculations, this is mainly due to the different metal coordination in the different MOFs, which may lead to different attack modes and power of NH_3_ to the metal center. In general, this study not only provides an important first step in designing suitable MOFs for the removal of NH_3_ from the composting process but also establishes a theoretical system for specific adsorption targets in the composting process.

## Figures and Tables

**Figure 1 molecules-27-07847-f001:**
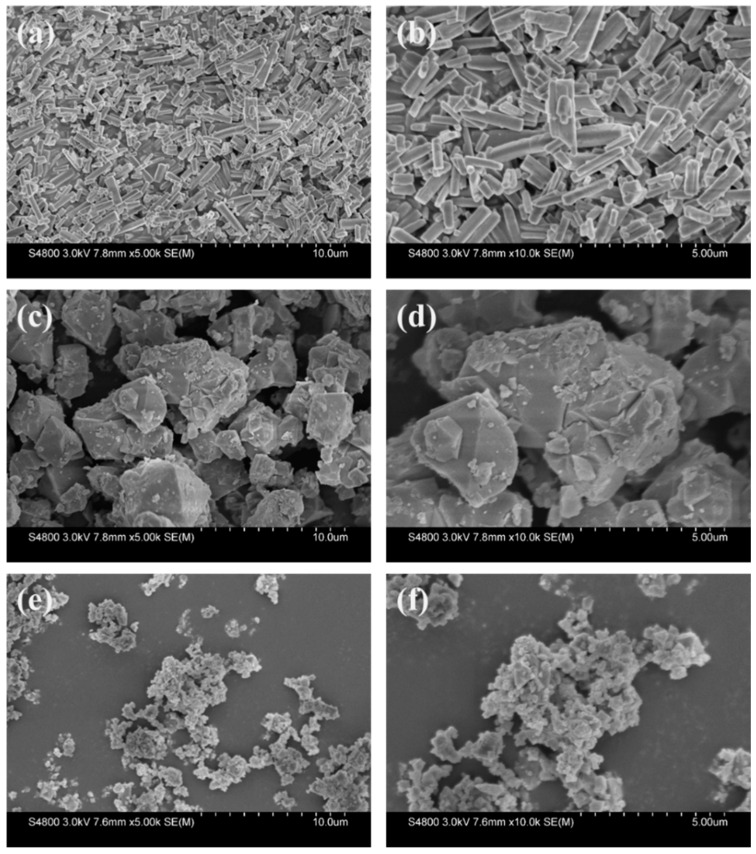
The SEM images of ZnBTC (**a**), the enlarged images of ZnBTC (**b**), FeBTC (**c**), FeBTC (**d**), CuBTC (**e**), and CuBTC (**f**).

**Figure 2 molecules-27-07847-f002:**
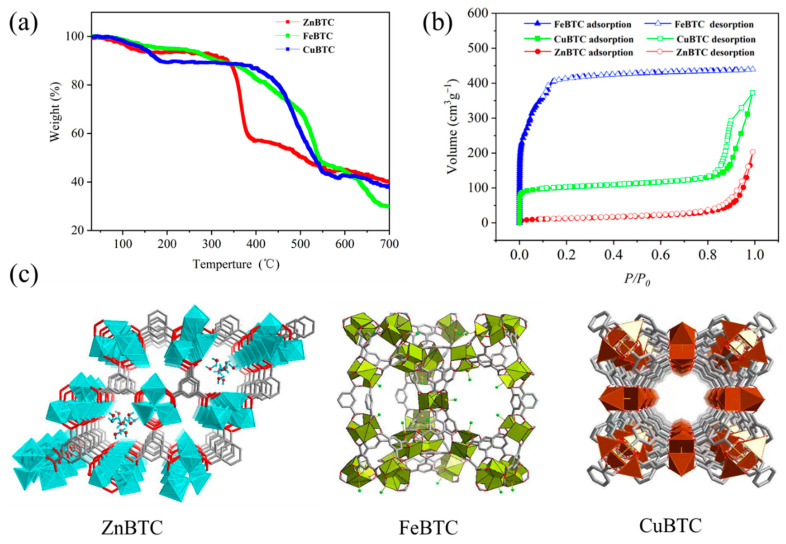
The TGA curve (**a**), N_2_ adsorption−desorption isotherms (**b**) and the crystal character of ZnBTC, FeBTC and CuBTC (**c**).

**Figure 3 molecules-27-07847-f003:**
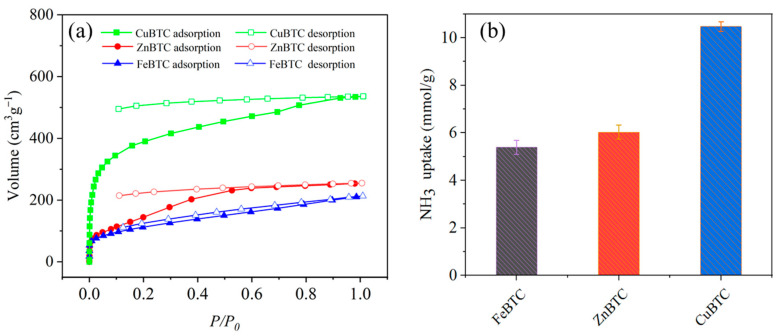
NH_3_ adsorption capacity of ZnBTC, FeBTC and CuBTC with different metal centers (**a**), and NH_3_ adsorption capacity of ZnBTC, FeBTC and CuBTC under an ammonia atmosphere (**b**).

**Figure 4 molecules-27-07847-f004:**
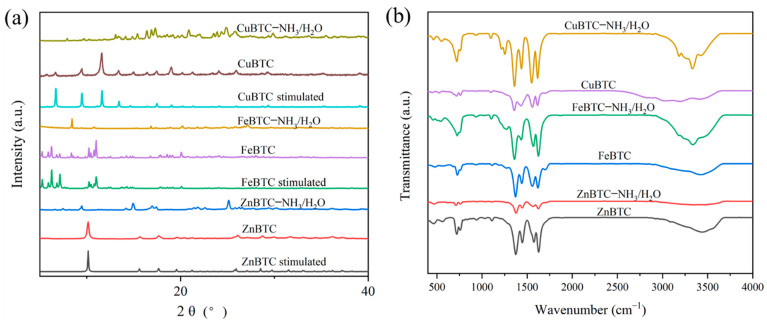
The XRD patterns (**a**) and FTIR (**b**) of ZnBTC, FeBTC and CuBTC before and after NH_3_ adsorption.

**Figure 5 molecules-27-07847-f005:**
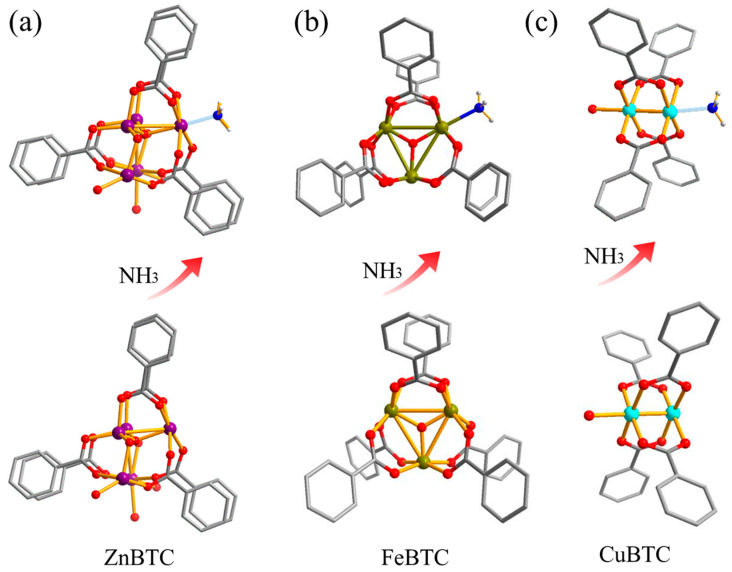
Adsorption configuration and distance of NH_3_ on metal site: (**a**) ZnBTC, (**b**) FeBTC and (**c**) CuBTC.

**Table 1 molecules-27-07847-t001:** The results of the theoretical calculation of Zn-BTC, FeBTC and CuBTC.

MOFs	Adsorbate	*E*_MOF(NH_3_)_ (kcal/mol)	*E*_MOF_ (kcal/mol)	*E*_NH_3__ (kcal/mol)	Δ*E* (kcal/mol)
CuBTC	NH_3_	−1,384,454.491	−1,348,955.985	−35,483.834	−14.7
FeBTC	NH_3_	−1,960,018.095	−1,924,515.523	−35,483.834	−18.7
ZnBTC	NH_3_	−2,158,051.865	−2,122,535.112	−35,483.834	−32.9

## Data Availability

All data generated or analyzed during this study are included in this published article. All data, models, and code generated or used during the study appear in the submitted article.

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
