# Peer review of "Tunable Ammonia Adsorption within Metal–Organic Frameworks with Different Unsaturated Metal Sites"

_molecules, 2022, doi:10.3390/molecules27227847_

Round 1

Reviewer 1 Report

This work by Zhang and colleages describes the employment of CuBTC, ZnBTC and ZnBTC MOFs in the adsorption of ammonia.

The research is well-conducted, but there are some points that the authors should address in any subsequent revision:

1. Elemental analysis of the pristine MOFs and the ammonia-loaded MOFs should be performed. A drastic increase in the nitrogen content should be observed.

2. The authors should explain general concepts of MOFs in the introduction part. 

3. The authors should highlight the novelty of their work. 

Author Response

Dear Editor and Reviewers,

Thanks very much for your critical comments for our manuscript. The opinions are very valuable for us to revise it further. Now, we have addressed your concerns point-by-point in the attached document as below, and the revised parts have been indicated with the red-colored words in the revised version of the manuscript.

*****Responses to the comments *****

Comments from the editors and reviewers:
-Reviewer 1
This work by Zhang and colleages describes the employment of CuBTC, ZnBTC and ZnBTC MOFs in the adsorption of ammonia.

The research is well-conducted, but there are some points that the authors should address in any subsequent revision:

  1. Elemental analysis of the pristine MOFs and the ammonia-loaded MOFs should be performed. A drastic increase in the nitrogen content should be observed.

Response: Thank you for your comments. According to your suggestion, elemental analysis of the pristine MOFs and the ammonia-loaded MOFs was performed. A drastic increase in the nitrogen content were observed which may attribute to the adsorption of NH3.

ZnBTC                             ZnBTC NH3            

FeBTC                             FeBTC NH3

CuBTC                            CuBTC NH3

  1. The authors should explain general concepts of MOFs in the introduction part.

Response: Thank you for your comments. The general concepts of MOFs have been added in the introduction part.

  1. The authors should highlight the novelty of their work.

Response: Thank you for your comments. The innovation of this work lies in the systematic analysis and comparison of the adsorption properties of three materials for the removal of ammonia gas.

Reviewer 2 Report

The presented research has been performed on a poor scientific level and is not possible to be published. 

1. "CuBTC" (HKUST-1) and "FeBTC" (Fe-MIL-100) have been studied in ammonia adsorption for many times [10.1002/advs.202002142]. Similar investigation of "ZnBTC" was already published recently by the authors [10.3390/molecules27175615]. Therefore, no novelty presents in this work. 

2. The samples are poorly characterized. No chemical composition and purity of all the samples are evidenced in fact. PXRD patterns are not compared to any theorectial one, and moreover, they do not match to the patterns of ZnBTC and CuBTC presented in the references, suggested by the authors. Zn- and Cu-based samples possess very low N2 uptakes and surface areas when compared to the literature data, what indicates that these MOF samples are impure. Thereefore, the investigation of NH3 adsorption properties for them is improper.  

3. No recyclability of Zn- and Cu- based samples in NH3 adsorption is shown, as the low-pressure region of the desorption isotherms (fig. 3a) is absent. PXRD data (fig. 4a) clearly indicate the decomposition of all three MOFs during the ammonia adsorption. 

4. Very low quality of the figures complicates their authenticity verification. 

Author Response

Dear Editor and Reviewers,

Thanks very much for your critical comments for our manuscript. The opinions are very valuable for us to revise it further. Now, we have addressed your concerns point-by-point in the attached document as below, and the revised parts have been indicated with the red-colored words in the revised version of the manuscript.

*****Responses to the comments *****

Comments from the editors and reviewers:

-Reviewer 2
The presented research has been performed on a poor scientific level and is not possible to be published.

  1. "CuBTC" (HKUST-1) and "FeBTC" (Fe-MIL-100) have been studied in ammonia adsorption for many times [10.1002/advs.202002142]. Similar investigation of "ZnBTC" was already published recently by the authors [10.3390/molecules27175615]. Therefore, no novelty presents in this work.

Response: Thank you for your comments. We agree with this reviewer’s comment that "CuBTC" (HKUST-1) and "FeBTC" (Fe-MIL-100) have been studied in ammonia adsorption for many times [10.1002/advs.202002142]. However, there are few mechanism researches and systematically compared about its adsorption properties of MOFs with different metal centers. In this study, we discuss the design of MOFs with different metals with count and theory. We can draw the conclusion: the adsorption capacity can be changed with the central metals.

  1. The samples are poorly characterized. No chemical composition and purity of all the samples are evidenced in fact. PXRD patterns are not compared to any theorectial one, and moreover, they do not match to the patterns of ZnBTC and CuBTC presented in the references, suggested by the authors. Zn- and Cu-based samples possess very low N2 uptakes and surface areas when compared to the literature data, what indicates that these MOF samples are impure. Therefore, the investigation of NH3 adsorption properties for them is improper.

Response: Thank you for your comments. It may be the way of mapping that causes the XRD data to appear inconsistent with those reported in the literature. According to the comments, the picture was remade. The reliability of the system was verified by comparing the results with those in the literatures. Many factors can affect the performance of N2 and NH3 uptakes of MOFs such as the purity of raw materials, proper production technologies and metal centers. Different preparation processes may lead to the very low N2 uptakes of Zn- and Cu-based samples. Hence, it is necessary to do this research about the preparation process and different metal centers.

  1. No recyclability of Zn- and Cu- based samples in NH3 adsorption is shown, as the low-pressure region of the desorption isotherms (fig. 3a) is absent. PXRD data (fig. 4a) clearly indicate the decomposition of all three MOFs during the ammonia adsorption.

Response: Thank you for your comments. As shown in the PXRD data, the decomposition of all three MOFs during the ammonia adsorption was obviously observed. Hence, no recyclability of Zn- and Cu- based samples in NH3 adsorption can be detected. 

  1. Very low quality of the figures complicates their authenticity verification.

Response: Thank you for your comments. We replaced some of the images to improve the quality of the figures.

Round 2

Reviewer 2 Report

Authors have not given any convenient explanations for the crucial faults, which were noticed in the first review, both in the revised version of the manuscript and in the authors response to the reviewer. In particular, if referring to the notes suggested in my first review: 

1. The suggested author's conclusion concerning the novelty of the presented work is a very trivial and also repeats many works mentioned in the first review. The theoretical calculations added into the present MS might be suitable if the structures of the synthesized materials were verified, but no any real structure confirmation exists in this work. 

2. No evidences of the phase purity and compositions of the synthesized materials (for example, by the apposition of PXRDs for the synthesized samples and theoretical PXRD patterns on the same graphic) are still included into the manuscript. Therefore, I can only repeat that gas adsorption investigations for these samples are fully meaningless. 

3. The authors response for this note confirms the reviewer's opinion from the previous note. If the sample decomposes at first cycle of ammonia adsorption, there is no any ground to confide matching between its speculated structure and the obtained adsorption data, as the structures and compositions of the phases which really adsorb NH3 are in fact unknown. 

Therefore, I can only regrettably confirm that the presented manuscript must be rejected due to the unresolvable faults in the characterization of the synthesized samples.